# Nursing Students’ Perception on the Effectiveness of Emergency Competence Learning through Simulation

**DOI:** 10.3390/healthcare8040397

**Published:** 2020-10-13

**Authors:** Ignacio Manuel Guerrero-Martínez, Francisco Javier Portero-Prados, Rocío Cándida Romero-González, Rocío Romero-Castillo, Manuel Pabón-Carrasco, José Antonio Ponce-Blandón

**Affiliations:** 1Sanitary Professional Training Center, SAMU Emergency School, Gelves, 41120 Seville, Spain; ignacioguerreromartinez@gmail.com; 2Red Cross Nursing School, University of Seville, 41009 Seville, Spain; jportero@cruzroja.es (F.J.P.-P.); rorogo@cruzroja.es (R.C.R.-G.); mpabon@cruzroja.es (M.P.-C.); japonce@cruzroja.es (J.A.P.-B.)

**Keywords:** simulation training, emergency nursing, simulation exercise, nurses

## Abstract

(1) *Background*: Simulation is a part of the day-to-day of the learning method in health sciences. The objective is to determine if the clinical simulation is useful for learning in the emergency setting, from the point of view of the nursing students. (2) *Methods*: A pre- and post-test exploratory study with an analytical and quasi-experimental design was used. The population is made up of nursing students from the Seville Red Cross Nursing Centre, who conducted a simulation exercise in the form of a drill for the care of multiple victims. A specific questionnaire was employed as a tool to analyse the dimensions of satisfaction, confidence and motivation, clinical experience, and decision making and technical abilities. (3) *Results*: There were favourable significant differences in the set of global responses, with *p* < 0.0001 for the “satisfaction” dimension and d = 1.25 for the “large” size of the effect, and *p* < 0.0069 for the “confidence and motivation” dimension and d = 0.58 for the “moderate–large” size of the effect. (4) *Conclusions*: The results are similar to those obtained in other studies in the scope of the 4 dimensions studied, thus coming to the conclusion that the perception of the nursing students on learning through clinical simulation is positive and favourable.

## 1. Introduction

The word “Simulate” comes from the Latin: *simulare*, and means “to represent something, by faking or imitating what that something is not” (Real Academia Española). However, in the setting of the clinical simulation, there is no agreed upon definition. For the Centre for Medical Simulation (Cambridge, Massachusetts), it is a situation or scenario created to allow people to experiment with the representation of a real event with the purpose of practicing, learning, evaluating, testing, or acquiring knowledge of human systems or actions [1].

Simulation is deeply introduced in health science education and is accepted as an educational method and as a tool which offers safety to the patient [2]. Apart from that, there are studies that show that the practice in groups focused on the patient, teamwork, and conflict resolution improves, increases, and promotes the care provided to the patient [3]. Some authors studied the perception of the nursing students in relation to the use of the clinical simulation as a learning strategy, thus demonstrating the importance of continuing the research in this field [4,5].

Summarising the relevant literature, studies are found like the “SIREN” (Simulation for emergency nurses) study, a quasi-experimental paper which asserts that simulation has been recognised as an effective learning strategy, improving patient safety and the clinical outcomes; however, it is necessary and important to evaluate the effectiveness of this educational technique to support its value and effectiveness [6]. Another study shows that the simulation helped the students to understand concepts and to stimulate critical thinking, as well as being a valuable and realistic learning experience [7]. Additionally, there are studies that identify the benefits of participating in the simulation, including the psychomotor abilities and those of evaluation, decision making, and critical thinking [8]. 

If we focus on the methodology and on the simulation exercises, different types can be found, among which the following stand out: simulation with actors (trained actors or simulated patients who reproduce interpersonal situations), simulation with hybrids (trained actors combined with an inanimate simulator for a realistic scenario and to streamline abilities), simulation with role games (based on scenic simulation), or e-training simulations (online) [9]. 

The purpose of this study can be defended and justified by referencing all these aforementioned studies and the interest in exploring a specific setting of the training in emergencies for the future nursing professionals in the application of these simulation techniques. On the one hand, it is claimed that, although the clinical simulation in nursing education is more and more frequent, the value of simulation in continuing education has not yet been accepted [10], and that, with the capacity to design scenarios with diverse clinical complexities, the simulation promotes permanent learning in the student [11]. 

In this way, the main objective of this study is to determine if the clinical simulation is useful for learning in the setting of nursing emergencies; thus, we set out the following hypotheses formulated in the form of research questions:
Is the emergency simulation activity satisfactory for the nursing students?Does the emergency simulation activity improve the students’ confidence and motivation?Does the emergency simulation activity improve the students’ perception of the clinical experience?Does the emergency simulation activity improve the students’ perception of the decision making and technical abilities?

## 2. Materials and Methods 

### 2.1. Study Design

An exploratory study of analytical and quasi-experimental design was set out through the distribution and completion of a questionnaire for the students both before and after an intervention (pre- and post-test), with the aim of studying the perception of the nursing students in a University Centre on learning through simulation, focusing in four blocks: satisfaction, confidence and motivation, clinical experience, and decision making and technical abilities.

### 2.2. Setting and Participants

The study population was made up of 51 students from the fourth year of the nursing graduate course at the Red Cross University Centre, attached to the University of Seville. The students had enrolled during the first four-month period of the 2018/2019 school year in the “Care for Multiple Victims and Humanitarian Help” discipline, with 4 being excluded for not expressing their will to participate, thus leaving a definite sample of 47 students. In order to make up the sample, the possibility of participating in the intervention was announced in the virtual learning platform of the discipline to all those enrolled in the aforementioned subject. Eventually, the pre-test was performed by the 47 students of the definitive initial sample, while 41 participated in the post-test, with a total of 6 being excluded from the analysis due to withdrawal. In this trial, investigators followed the rules of the CONSORT (Consolidated Standards of Reporting Trials) declaration [12].

The intervention was structured into an exercise with two drills executed on 29 November 2018, in such a way that the first one consisted in a simulation of a huge explosion, apparently of a terrorist nature, in a building of the autonomous administration of Andalusia (1 h and 30 min in full), and the second one, in a collision in which several vehicles were involved (1 h and 30 min in full). In the first exercise, there were 16 victims, and 11 victims in the second exercise. The students participated in a briefing session, where they were also given descriptive documentation of both drills with the corresponding role assignments for each one of the participants, handing them all the necessary material for their development. Once the drills were over, a reflexive briefing session was conducted on the learned aspects and on the vulnerable items to improve the clinical practice in emergencies.

In relation to the study variables, the only demographic variables collected were gender and age. The main study variables were realized in four groups, corresponding to the dimensions of the data collection tool: “satisfaction”, “confidence and motivation”, “clinical experience”, and “decision making and technical abilities”.

### 2.3. Data Collection

For data collection, a specific questionnaire was used on the perception of learning through simulation, configured with 11 items for the first block, 10 items for the second, 7 items for the third, and 14 items for the fourth. It was to be answered according to the level of agreement or disagreement with an assertion by means of a 5-point Likert scale, where the value of 1 corresponds to “completely disagree” and the value of 5 means “completely agree”. Such a tool is a questionnaire adapted from the quality and satisfaction survey of clinical simulation answered by nursing students in an extended version, conducted through a committee of experts in the “Care for Multiple Victims and Humanitarian Help” discipline from the Red Cross University Centre, attached to the University of Seville. The original version consisted of 15 items distributed in 3 dimensions, with a global Cronbach’s alpha of 0.861 [13].

The pre-test questionnaire was distributed in person one week before the intervention, in a meeting held with the participants for the briefing of the session, while the post-test questionnaire was distributed also in person after the meeting, in the final debriefing.

### 2.4. Data Analysis

All the data were poured into a database constructed in the Epi Info application, version 7.2.3.1 (Centers for Disease Control and Prevention, Atlanta, GA, USA), with its data analysis tool being employed to conduct the descriptive analysis and to contrast all the hypotheses. The questionnaires were on paper. They were completed individually by students. A blind researcher poured the data into a computer. Another blind researcher reviewed this database. These researchers were unaware of the study objectives.

For the numerical scores of each item, the measures of central tendency and dispersion were calculated, both for each item and for the groupings of items by dimensions. The index of the Cohen’s d effect was calculated by comparing the initial mean scores with the final ones and by relating them to the standard deviation of the initial values. A “small” size was considered to be below 0.2, “small–moderate”, between 0.2 and 0.5, “moderate–large”, between 0.51 and 0.79, and “large”, over 0.79 [14].

In order to contrast the hypotheses, the Student’s t test for the comparison of two measures for paired data was used. The data normality conditions were previously verified by means of the Shapiro–Wilk test so as to be able to apply the test. The significance level was established as a *p*-value < 0.05.

As regards the ethical implications of the study, the confidentiality of the handled information was preserved, including information on the informed consent and the anonymity guarantee in the questionnaire, so that those students who did not sign this consent were excluded. The research commission of the centre granted the necessary authorisation.

## 3. Results

Forty-seven students participated in the pre-test, and forty-one in the post-test. Of the 47 initial students, 17.02% (*n* = 8) were male and 82.98% (*n* = 39) were women. The mean age was 21.5 years old (standard deviation (SD) = 2.6). 

Table 1, Table 2, Table 3 and Table 4 reproduce the global results obtained in the pre-test and in the post-test in the four dimensions of the questionnaire and, in each of its items, their level of significance and size of the effect by comparing the valuation mean values and by incorporating the result of the hypothesis test.

### 3.1. Satisfaction

Favourable significant differences were found in the global score (*p* < 0.0001) and in all the items, except for 3 of them: “I think that developing activities in the simulation allows me to enrich my knowledge from the experience”, “I think that the teacher accompanying me in the simulation improves my learning”, and “I consider that the simulation I did is enough for my learning”. In those items, although the differences were not significant, all the post-test mean values were higher to their pre-test counterparts. The size of the effect on the global score was “large” (d = 1.25), as well as in the majority of the items with significant differences, which indicated an improvement in the score (Table 1).

### 3.2. Confidence and Motivation

Favourable significant differences were again found in the global score (*p* < 0.0069) and in most of the items, except for item 4, although, similarly to the previous dimension, these items showed an increased mean value in the post-test even if the result is not significant. The size of the effect both in the global score (d = 0.58) and in most of the items with significant differences was “moderate–large” (Table 2).

### 3.3. Perception of the Clinical Experience

Both the global score (*p* < 0.0002) and all the items from this dimension presented favourable significant differences. The size of the effect in the global score was “large” (d = 1), with all of its items presenting a “moderate–large” and a “large” size of the effect (Table 3).

### 3.4. Perception of the Decision Making and Technical Abilities

Just as in the previous dimension, favourable significant differences were obtained in all the scores (*p* < 0.0127). The size of the effect in the global score was “moderate–large” (d = 0.6), as well as in all its items, except for item number 7, in which a “large” size of the effect was obtained (d = 0.84) (Table 4).

## 4. Discussion

After the detailed and careful analysis of each of the questionnaire items in question, and in view of the results, an improvement is appreciated in the scores of all the dimensions, which leads to considering that all the research questions set out have an affirmative answer. This can be objectivised by valuing the sizes of the Cohen’s d effect and the statistical significance of each of the items and dimensions. In relation to the size of the effect observed, it is worth highlighting that, for the globality of the 4 dimensions mainly studied, the figures obtained according to the reference values [14] have turned out to be “moderate–large” (“confidence and motivation” and “perception of the decision making and technical abilities”) and “large” (“satisfaction” and “perception of the clinical experience”), just as for the majority of the specific items. Additionally, in the 4 dimensions studied, there are favourable significant differences.

With regard to the methodological questions, although the tool used for this study of an exploratory character is adapted from a validated tool, it can be perceived as a new tool expanded from the aforementioned one which, in a future, could set itself out to design specific research papers to validate the adaptation of the “Survey on the quality and satisfaction of the clinical simulation for nursing students” [13]. 

When conducting the comparison of the results obtained with those of other similar questionnaires and studies, generalised similitudes can be seen, as well as that the group under study has expressed similar answers to those of other students. 

One of these studies concludes that the simulation improves learning and retention of the acquired knowledge, as well as that it is a useful methodology to improve the learning model. This is also reflected in the “perception of the clinical experience” and “perception of the decision making and technical abilities” dimensions of this study. The study also asserts that the students who conducted simulation sessions express a high level of satisfaction, a fact also comparable to our findings [9]. 

Another study asserts that the simulation reduces the participants’ anxiety and improves self-efficacy in the evaluation of the patient and that, apart from that, the nurses were very satisfied with the simulation training, also coinciding with the results obtained in this study [6]. Still in the same line, another of these studies reconfirms the results obtained herein, where statements like “The simulation is a useful teaching method for learning” or “The simulation has helped me to integrate theory and practice” obtain mean values of 4.7 and 4.2 respectively, in a Likert scale where 5 is the highest score [13].

Another study which also analyses the 4 dimensions studied herein, but which uses a semi-structured interview as its method, arrives to the same results to ours, and concludes that the participants improve their learning and knowledge with the simulation, get closer to the real practice field, acquire confidence for their clinical and professional practice, and feel satisfied with the use of the clinical simulation [15]. 

Another study conducted in the nursing graduate course from the UCAM (Universidad Católica San Antonio de Murcia), with the objective of knowing the perception and opinion of its students on the clinical simulation, concluded that such perception was positive, specifically valuing the acquisition of competences (prioritisation, knowledge reinforcement, correction of errors, previous training to the real practice) and showed that, once again, the results obtained herein are similar and arrive at the same conclusion [16]. 

A study conducted in the Autonomous University of Carmen (Mexico), where the students’ perception of their satisfaction with the clinical simulation as a teaching and learning technique is determined, arrived at results also similar to those of our study, thus asserting that the clinical simulation is an excellent learning strategy which allows them to integrate theory and practice, as well as to improve critical thinking, reinforce knowledge, abilities, skills, perception of the decision making, and professional ethics [17] 

In this way, after the analysis and subsequent comparison of similar studies, it is verified that the results obtained are similar and favourable in terms of the students’ perception on the clinical simulation; thus, it is proposed to continue with this type of intervention in order to further improve the level of satisfaction, confidence and motivation, perception of the clinical experience, and decision making and technical abilities.

### Limitations

In relation to the methodological aspects, the main limitation to be highlighted is that, although the data collection tool was adapted from a previously validated questionnaire, such adaptation has not been validated in its final version, apart from not being designed for pre- and post-tests. Even so, however, it is useful for the purpose of the study according to the indications of the panel of subject matter experts which made those adaptations, since this is an exploratory and complete study as regards the dimensions measured. Additionally, as already mentioned in the “Methods” section, the original “Survey on the quality and satisfaction of the clinical simulation for nursing students” validated tool had a high level of reliability [13], which could be extrapolated to the tool employed.

## 5. Conclusions

The results of this study showed preliminary data from the simulation in group of students, showing that the clinical simulation in the emergency setting generates satisfaction, confidence, and motivation in the students.

It is necessary to improve and adapt the available tools for measuring its effectiveness, conducting specific research studies to verify its validity and reliability, and so that it can be used as an optimal tool to continue the study of the students’ perception on the simulation in nursing emergencies and its efficacy in the acquisition of clinical competences.

In view of all this, even being aware of the aforementioned limitations, it can be said that this study has a positive relevance in the scope of the clinical simulation.

## Figures and Tables

**Table 1 healthcare-08-00397-t001:** Comparisons of the mean scores obtained in the pre-test and in the post-test. Module on satisfaction.

Questionnaire Items	Pre-Test (Mean; SD)		Post-Test (Mean; SD)		Student’s *t* Value; DoF; *p*-Value	Size of the Cohen’s d Effect
1. I consider that the simulation experience has prepared me adequately.	3.48	0.77	4.36	0.58	*t* = 5.92; DoF = 86; *p* < 0.0001 *	1.14
2. The allotted time for the simulation was adequate.	3.42	0.94	4.36	0.66	*t* = 5.30; DoF = 86; *p* < 0.0001	1.00
3. I am satisfied with the simulation experience.	3.93	0.84	4.85	0.42	*t* = 6.30; DoF = 86; *p* < 0.0001	1.10
4. In general, the simulation experience improved my learning.	4.29	0.62	4.60	0.58	*t* = 2.41; DoF = 86; *p* = 0.0182	0.50
5. I consider that the physical space in which the simulation was developed facilitates its development.	3.51	1.03	4.82	0.49	*t* = 7.42; DoF = 86; *p* < 0.0001	1.27
6. I believe that the exercise had enough simulation elements for me to learn.	3.91	0.81	4.65	0.43	*t* = 4.36; DoF = 86; *p* < 0.0001	0.91
7. I think that the time has been enough for me to practice.	3.29	0.97	4.46	0.55	*t* = 6.76; DoF = 86; *p* < 0.0001	1.21
8. I consider that the development of the simulation complements what was learned in class.	4.38	0.79	4.73	0.50	*t* = 6.30; DoF = 86; *p* < 0.0001	0.44
9. I think that developing activities in simulations allows me to enrich my knowledge from the experience.	4.55	0.74	4.68	0.68	*t* = 0.84; DoF = 86; *p* = 0.401	0.18
10. I believe that the teacher accompanying me in the simulation improves my learning.	4.70	0.58	4.75	0.53	*t* = 0.45; DoF = 86; *p* = 0.655	0.09
11. I consider that the simulation I did was enough for my learning.	3.29	1.06	3.65	0.91	*t* = 1.70; DoF = 86; *p* = 0.093	0.34
Mean scores of the satisfaction module	3.89	0.52	4.54	0.27	*t* = 7.08; DoF = 86; *p* < 0.0001	1.25

SD: Standard deviation; DoF: Degrees of freedom; *: Statistically significant results.

**Table 2 healthcare-08-00397-t002:** Comparisons of the mean scores obtained in the pre-test and in the post-test. Module on confidence and motivation.

Questionnaire Items	Pre-Test (Mean; SD)		Post-Test (Mean; SD)		Student’s *t* Value; DoF; *p*-Value	Size of the Cohen’s d Effect
1. My experience with the simulation increased my level of confidence to face the real setting.	3.63	0.94	4.07	0.84	*t* = 2.26; DoF = 86; *p* = 0.0262	0.47
2. Conducting the simulation motivated me to learn.	4.23	0.94	4.34	0.66	*t* = 0.71; DoF = 86; *p* = 0.482	0.12
3. The simulation gave me confidence in my technical abilities.	3.65	0.89	4.12	0.84	*t* = 2.49; DoF = 86; *p* = 0.0147	0.53
4. I consider that, if a teacher accompanies me during the simulation, I further develop my technical abilities.	4.17	0.98	4.31	1.01	*t* = 0.69; DoF = 86; *p* = 0.492	0.14
5. I consider that the teachers foster the simulation to improve my learning.	4.56	0.71	4.19	0.67	*t* = 2.49; DoF = 86; *p* = 0.0146	0.52
6. I believe that the practical activities in simulation increase my level of confidence.	4.03	1.02	4.53	0.83	*t* = 2.46; DoF = 86; *p* = 0.0160	0.49
7. I consider that the practical activities in simulation reduce my level of anxiety.	3.40	1.19	3.95	0.94	*t* = 2.36; DoF = 86; *p* < 0.0206	0.46
8. I am free to attend the development of the simulation.	3.31	1.56	3.53	1.48	*t* = 0.67; DoF = 86; *p* = 0.506	0.14
9. I feel forced to do the simulation exercise.	3.21	1.58	3.42	1.38	*t* = −0.64; DoF = 86; *p* = 0.521	0.13
10. I easily recognise the objectives of the simulation and the reasons to conduct it.	4.10	0.86	4.60	0.58	*t* = 3.15; DoF = 86; *p* = 0.0023	0.58
Mean scores of the confidence and motivation module	3.81	0.53	4.12	0.50	*t* = 2.77; DoF = 86; *p* = 0.0069	0.58

SD: Standard deviation; DoF: Degrees of freedom.

**Table 3 healthcare-08-00397-t003:** Comparisons of the mean scores obtained in the pre-test and in the post-test. Module on perception of the clinical experience.

Questionnaire Items	Pre-Test (Mean; SD)		Post-Test (Mean; SD)		Student’s *t* Value; DoF; *p*-Value	Size of the Cohen’s d Effect
1. The simulation is a realistic tool to learn to evaluate the real situation.	4.31	0.75	4.77	0.42	*t* = 3.50; DoF = 86; *p* = 0.0007	0.61
2. The scenarios used with the simulation recreate real-life situations.	3.93	0.89	4.75	0.54	*t* = 5.02; DoF = 86; *p* < 0.0001	0.92
3. The simulation scenario was realistic.	3.51	1.10	4.75	0.54	*t* = 6.48; DoF = 85; *p* < 0.0001	1.13
4. The pace of the simulation reflected the flow in a real setting.	3.14	1.08	4.27	0.93	*t* = 5.15; DoF = 85; *p* < 0.0001	1.05
5. I believe the simulation is a useful learning strategy to come closer to the challenges of the real practice.	4.21	0.95	4.75	0.43	*t* = 3.38; DoF = 85; *p* = 0.0015	0.57
6. I consider that the simulation allows me to learn in a realistic context which mimics the care provided to the patient.	4.10	0.81	4.60	0.70	*t* = 2.99; DoF = 85; *p* = 0.0037	0.62
7. I believe that the simulation mimics the care provided to the patient in a safe and controlled setting.	3.93	0.94	4.47	0.78	*t* = 2.87; DoF = 85; *p* = 0.0052	0.57
Mean scores of the clinical experience module	3.88	0.63	4.51	0.85	*t* = 3.95; DoF = 86; *p* = 0.0002	1.00

SD: Standard deviation; DoF: Degrees of freedom.

**Table 4 healthcare-08-00397-t004:** Comparisons of the mean scores obtained in the pre-test and in the post-test. Module on perception of the decision making and technical abilities.

Questionnaire Items	Pre-Test (Mean; SD)		Post-Test (Mean; SD)		Student’s *t* Value; DoF; *p*-Value	Size of the Cohen’s d Effect
1. The simulation is a realistic tool to learn to evaluate the patient in a real setting.	4.14	0.80	4.55	0.59	*t* = 2.60; DoF = 85; *p* = 0.0111	0.51
2. My experience with the simulation improved my technical abilities.	3.76	0.93	4.27	0.59	*t* = 2.96; DoF = 85; *p* = 0.0040	0.55
3. The scenarios develop critical thinking and decision making.	3.91	0.92	4.42	0.71	*t* = 2.84; DoF = 85; *p* = 0.0057	0.55
4. The prioritisation abilities taught by using the simulation are adequate.	4.00	0.80	4.55	0.59	*t* = 3.56; DoF = 85; *p* = 0.0006	0.69
5. My interaction with the simulation improved my clinical competence.	3.80	1.03	4.35	0.62	*t* = 2.89; DoF = 85; *p* = 0.0049	0.53
6. The simulation allowed me to put theory into practice.	4.12	0.99	4.60	0.49	*t* = 2.73; DoF = 85; *p* < 0.0076	0.48
7. The experiences with the simulation helped me to determine priority aspects in the nursing care.	3.85	0.80	4.52	0.59	*t* = 4.36; DoF = 85; *p* < 0.0001	0.84
8. The simulation helped me handle the clinical emergencies effectively.	3.63	0.98	4.27	0.67	*t* = 3.44; DoF = 85; *p* = 0.0009	0.65
9. My experience in the simulation gave me confidence in my technical abilities.	3.57	1.05	4.25	0.80	*t* = 3.30; DoF = 85; *p* = 0.0014	0.65
10. I consider that the practical activities developed in the simulation are significant for the development of technical abilities.	4.04	0.93	4.57	0.54	*t* = 3.27; DoF = 85; *p* = 0.0021	0.57
11. I consider that repeating actions in the simulation hones my technique to handle the patient.	4.38	0.87	4.72	0.55	*t* = 2.14; DoF = 85; *p* = 0.0355	0.39
12. The simulation improves my ability and capacity to apply my knowledge in different situations.	4.14	0.80	4.52	0.55	*t* = 2.49; DoF = 85; *p* = 0.0148	0.48
13. I consider that the simulation allows me to make decisions on the care provided to the patient.	4.00	0.85	4.35	0.73	*t* = 2.02; DoF = 85; *p* = 0.0464	0.41
14. The clinical simulation allowed me to develop abilities in the assertive communication with the multidisciplinary team.	3.93	0.81	4.52	0.50	*t* = 3.95; DoF = 85; *p* = 0.0002	0.73
Mean scores of the decision making and technical abilities module	3.95	0.67	4.35	0.81	*t* = 2.54; DoF = 86; *p* = 0.0127	0.60

SD: Standard deviation; DoF: Degrees of freedom.

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
