# Peer review of "Nursing Students’ Perception on the Effectiveness of Emergency Competence Learning through Simulation"

_healthcare, 2020, doi:10.3390/healthcare8040397_

Round 1

Reviewer 1 Report

Dear authors,  very compliments for these data. Please could you describe the role of simulation in the maxi emergency context for Nursing students? Thank you for this opportunity.

The authors describe that students participated in a briefing session, where they were also descriptive documentation of both drills with the corresponding role assignments for each one of the participants, handing them all the necessary material for their development. Once the drills were over, a reflexive briefing session was conducted on the learned aspects and on the vulnerable items to improve the clinical practice in emergencies. Please the authors can describe the chatacteristics of briefing session? For example I am following a master's course in disaster management at the University of Naples, with two theoretical lessons per week and a 1-week practical simulation exercise on an example of a maxi-emergency.
Please the authors can they better describe what it consists of?

Reviewer 2 Report

Very nice work on this interesting manuscript. You clearly defined a gap in nursing education regarding simulation exercises for mass casualty events. You may want to consider adding citations that show the use of mass casualty simulations among healthcare and government organizations, which also shows a logical progression to nursing students.

Just a few questions/comments regarding the methodology: 

Participants: it would helpful to see the demographics on these participants (i.e, age, gender, etc.)

Additional details on the setting: Can you briefly give more details on the simulations? How many exercise participants? How many victims in each exercise? Etc.

Data collection: Were the questionnaires on paper or electronic? Who 'poured in the the database'? If they were manually entered, did a second person double check the data entry process?

I am struggling a bit with some of the questions on the questionnaire and how they actually measure the research questions.

  1. The research question for clinical experience states "Does the emergency simulation activity improve the student's clinical experience?" Improve how? Compared to what? The "Clinical experience" questions focus on comparing the simulation to a real-life emergency and appear to assume that the nursing students have experience with real-life emergency situations. Does a nursing student really know if the simulation mimics the care provided to the patient or matches the pace of a real-life event? If the students have not been involved in a real-life emergency, I am not sure how they could correctly answer these questions. Also, I am missing the link between the research question and the clinical experience questions as I feel they are mismatched. In your title you mention the students' perception, yet the research questions do not include perception. I think it might be a better fit if your question was 'Does the emergency simulation activity improve the student's perception of the clinical experience?'
  2. "Does the emergency simulation activity improve the student’s decision making and technical abilities?" Again, the questions are not actually measuring or evaluating the student's decision making and technical abilities. You are only measuring the students' perception. I would suggest you re-frame the question to reflect this.
  3. "...apart from improving their clinical experience and their decision making and technical abilities." So you actually measured their perception of these items. You have not provided any evidence to show it improved their decision making and technical abilities.
  4. Limitations: You mention the original tool was validated. However, you adapted it and changed questions. To say the study used a validated tool maybe a slight overstatement. Also, you tested the tool on 41 students, which is a limited population. I am not sure your outcomes show "The results of this study clearly show through evidence that both the general objective and the four research questions set out obtain a favourable answer..." I definitely think this is a first step. But you would need to test the questionnaire in other nursing student populations (i.e., other schools, countries, etc.); and ideally, you would want a sufficient population to conduct a factor analysis to provide additional validation.

Very creative work and look forward to seeing more on the validation of the tool in the future!

Round 2

Reviewer 1 Report

Dear Authors Thank you for your response. For me it is accept. Best regards 

Reviewer 2 Report

Thank you for the opportunity to review this novel article that advances the use of simulation in nursing education specifically for emergencies. Very nice job in addressing the feedback and comments from the first review. Well done. Just a few punctuation errors that can be corrected during the editing process (e.g. Page 9, line 249 needs a period not a comma).

Looking forward to seeing this in print and seeing additional work in this area.